# Synthesis and systematic review of reported neonatal SARS-CoV-2 infections

Roberto Raschetti[1,5], Alexandre J. Vivanti [2,5], Christelle Vauloup-Fellous[3], Barbara Loi[1], Alexandra Benachi [2] & Daniele De Luca [1,4✉]

A number of severe acute respiratory syndrome coronavirus-2 (SARS-CoV-2) infections have been reported in neonates. Here, we aim to clarify the transmission route, clinical features and outcomes of these infections. We present a meta-analysis of 176 published cases of neonatal SARS-CoV-2 infections that were defined by at least one positive nasopharyngeal swab and/or the presence of specific IgM. We report that 70% and 30% of infections are due to environmental and vertical transmission, respectively. Our analysis shows that 55% of infected neonates developed COVID-19; the most common symptoms were fever (44%), gastrointestinal (36%), respiratory (52%) and neurological manifestations (18%), and lung imaging was abnormal in 64% of cases. A lack of mother–neonate separation from birth is associated with late SARS-CoV-2 infection (OR 4.94 (95% CI: 1.98–13.08), $p = 0.0002$; adjusted OR 6.6 (95% CI: 2.6–16), $p < 0.0001$), while breastfeeding is not (OR 0.35 (95% CI: 0.09–1.18), $p = 0.10$; adjusted OR 2.2 (95% CI: 0.7–6.5), $p = 0.148$). Our findings add to the literature on neonatal SARS-CoV-2 infections.

[1] Division of Pediatrics and Neonatal Critical Care, "A.Béclère" Medical Centre, Paris Saclay University Hospitals, APHP, Clamart, France. [2] Division of Obstetrics and Gynecology, "A.Béclère" Medical Centre, Paris Saclay University Hospitals, APHP, Clamart, France. [3] Division of Virology, "Paul Brousse" Hospital, Paris Saclay University Hospitals, APHP, Villejuif, France. [4] Physiopathology and Therapeutic Innovation Unit-INSERM, Paris Saclay University, U999 Le Plessis Robinson, France. [5]These authors contributed equally: Roberto Raschetti, Alexandre J. Vivanti. ✉email: dm.deluca@icloud.com

Since December 2019, the infection of a novel beta-coronavirus, the severe acute respiratory syndrome coronavirus-2 (SARS-CoV-2), has spread worldwide and caused a potentially lethal illness: the coronavirus disease 2019 (COVID-19). The World Health Organization (WHO) on 11 March 2020[1] declared the outbreak a public health emergency of international concern and, since then, ~27,000,000 cases have occurred worldwide (https://www.who.int/docs/default-source/coronavirus/situation-reports/20200907-weekly-epi-update-4.pdf?sfvrsn=f5f607ee_2). Of these cases ~40–45% is reported to be asymptomatic[2,3], but more than 900,000 infected patients have died (https://www.who.int/docs/default-source/coronavirus/situation-reports/20200907-weekly-epi-update-4.pdf?sfvrsn=f5f607ee_2).

COVID-19 may appear with a panoply of clinical manifestations. It usually starts with influenza-like symptoms, eventually progressing toward pneumonia and acute respiratory distress syndrome (ARDS), but cardiovascular, renal, neurological, dermatological, and gastrointestinal manifestations have also been reported[4,5]. The pathophysiology of COVID-19 includes the direct viral cytopathic effect, but also an exaggerated inflammatory response (the so-called "cytokine storm"), with hypercoagulable state and tissue damage[6]. COVID-19 is more frequent and tends to be more severe with increasing age and in patients with certain characteristics, some of which are represented by comorbidities typical of older adults (such as cardiovascular disorders and diabetes)[7,8]. These risk factors have a role both in the disease occurrence and in the progression towards greater clinical severity and mortality[7,8]. Moreover, children might be less prone to develop the chaotic inflammatory host response that contributes to the clinical picture of COVID-19[9] and the disease seems to have a milder clinical course in children than in adults[10]. However, the proportion of asymptomatic infections seems to be lower in children than in adults[11,12].

Despite these findings, the knowledge about pediatric SARS-CoV-2 infection is still based on a paucity of data and the maternal-fetal transmission of SARS-CoV-2 was initially deemed uncertain. However, with the pandemic spreading around the world, some cases of SARS-CoV-2 infection in the first month of life have been reported in the literature or in the mass media[9], and it has been hypothesized that neonatal COVID-19 might occasionally become clinically apparent. SARS-CoV-2 has been isolated in placental tissues[13,14] and, in a few cases, maternal–fetal transmission was suspected[15]. Nonetheless, guidance criteria for the diagnosis of perinatal and neonatal SARS-CoV-2 infections have been released only recently and the cases published so far were not systematically analyzed according to these criteria[16].

Compared with adult medicine, neonatology suffers from a clear knowledge gap about the SARS-CoV-2 infection. In fact, the main transmission route is unknown, as neonates can be infected antenatally (through the placenta), during the delivery, or postnatally (through environmental exposure) and the clinical features of neonatal SARS-CoV-2 infections are unclear. We performed a systematic review and synthesis of published cases of neonatal SARS-CoV-2 infections, having as primary objective to describe the route of transmission and the clinical characteristics of neonatal SARS-CoV-2 infections. Our secondary objective was to clarify the effect of mother-neonate separation and breastfeeding on the incidence of late-onset neonatal infections.

## Results

**Workflow of review, synthesis, and meta-analysis.** Figure 1 illustrates the project flow chart with included and excluded records (and the reasons for their exclusions). Finally, 74 articles were considered, consisting of 37 case series, 34 case reports, two retrospective cohort studies, and one cross-sectional study[17–90].

Only three articles were not peer-reviewed already. The paper characteristics are reported in Table 1: the methodological quality of case reports and series was estimated as intermediate (median score 5 [2;6]).

**Basic patients' characteristics.** These articles described 176 neonates infected by SARS-CoV-2, whose general characteristics are reported in Table 2. Only two neonatal infections were diagnosed by the presence of IgM in newborn blood[29,82]: these cases qualified as possible congenital and probable postpartum acquired infections, respectively, according to the classification system[16]; all the other cases had at least one positive RT-PCR. Nine (5.1%) neonates required delivery room resuscitation[19,20,38,41,44,69,77,87] and 67 (38.3%) needed to be admitted to the NICU[17,18,20,27,31,33,35,36,38,41,44,45,47–49,51–53,57–59,61,62,64–66,69–73,77,80–82,85–88]. The median NICU stay was 8 [5;18] days (min 1; max 69 days). Of these neonates, five were extremely preterm (of 26, 27 and 28 weeks' gestation; birth weight ranging between 900 and 1410 g, respectively), five were preterm (31 and 32 weeks' gestation; birth weight ranging between 1580 and 2300 g), nine were moderately preterm (33 and 34 weeks' gestation; birth weight ranging between 1839 and 3280 g) and eleven were late preterm (35 and 36 weeks' gestation; birth weight ranging between 2000 and 2540 g); all other NICU-admitted neonates were term infants.

**Transmission routes of neonatal SARS-CoV-2 infections.** SARS-CoV-2 infection was classified[16] as shown in Fig. 2: the majority of infections were likely transmitted postpartum, that is, they were due to environmental exposure, although ~30% of cases were likely due to vertical transmission, either intrapartum or congenitally. Of these, ~9% of cases were confirmed vertical infections (3.3% for intrapartum and 5.7% for congenitally transmitted infections, respectively).

**Clinical features of neonatal SARS-CoV-2 infections.** Ninety-seven neonates presented with clinical features related to COVID-19[18,20,26–28,30,32,34–36,38–41,44,45,50–52,55,56,58,60,61,65–67,70–72,77,78,80,81,85–88,90] their distribution is shown in Table 3. Respiratory manifestations mainly consisted of signs of respiratory failure, such as tachypnoea, intercostal retractions, and rhinitis; neonatal ARDS was not diagnosed in any case. Gastrointestinal manifestations were primarily represented by feeding difficulties, diarrhea, and vomiting, while neurological ones consisted of hypertonia and irritability, but also hypotonia and lethargy, as well as apnea. Cardiovascular features were tachycardia and hypotension. Other manifestations included conjunctivitis, hypothermia and cutaneous rash. Laboratory abnormalities were evident in a minority of cases: 14 (14.4%) and 4 (4.1%) out of 97 neonates presented with lymphopenia[20,41,53,55,59,64,69,72,79–81,88] and raised liver enzymes[35,72,79,88], respectively. Inflammatory markers (C-reactive protein and procalcitonin) were increased in 15 (15.5%) out of 97 neonates[20,27,38,40,43,45,49,59,70,72,80–82,88]. Lung imaging was abnormal in 62 (64%) out of 97 neonates and consisted of an interstitial-alveolar pattern at lung ultrasound or chest X-ray and ground-glass opacities at CT-scan[17,19,21,23,27,29,32,36,38,44,45,51,53,55,58,59,64–66,69–71,73,77–80,82,85,88,90]. One neonate presenting with neurological manifestations also showed bilateral gliosis of the deep white periventricular and subcortical matter, together with signs of cerebral vasculitis, which was not totally remitted at the hospital discharge[77]. Seven neonates were treated with oral hydroxychloroquine and/or azithromycin, two with intranasal interferon-α1b and one with intravenous remdesivir, oseltamivir and ritonavir/lopinavir, respectively[17,27,36,45,49,53,59,69,80,82,85]; all the others received only

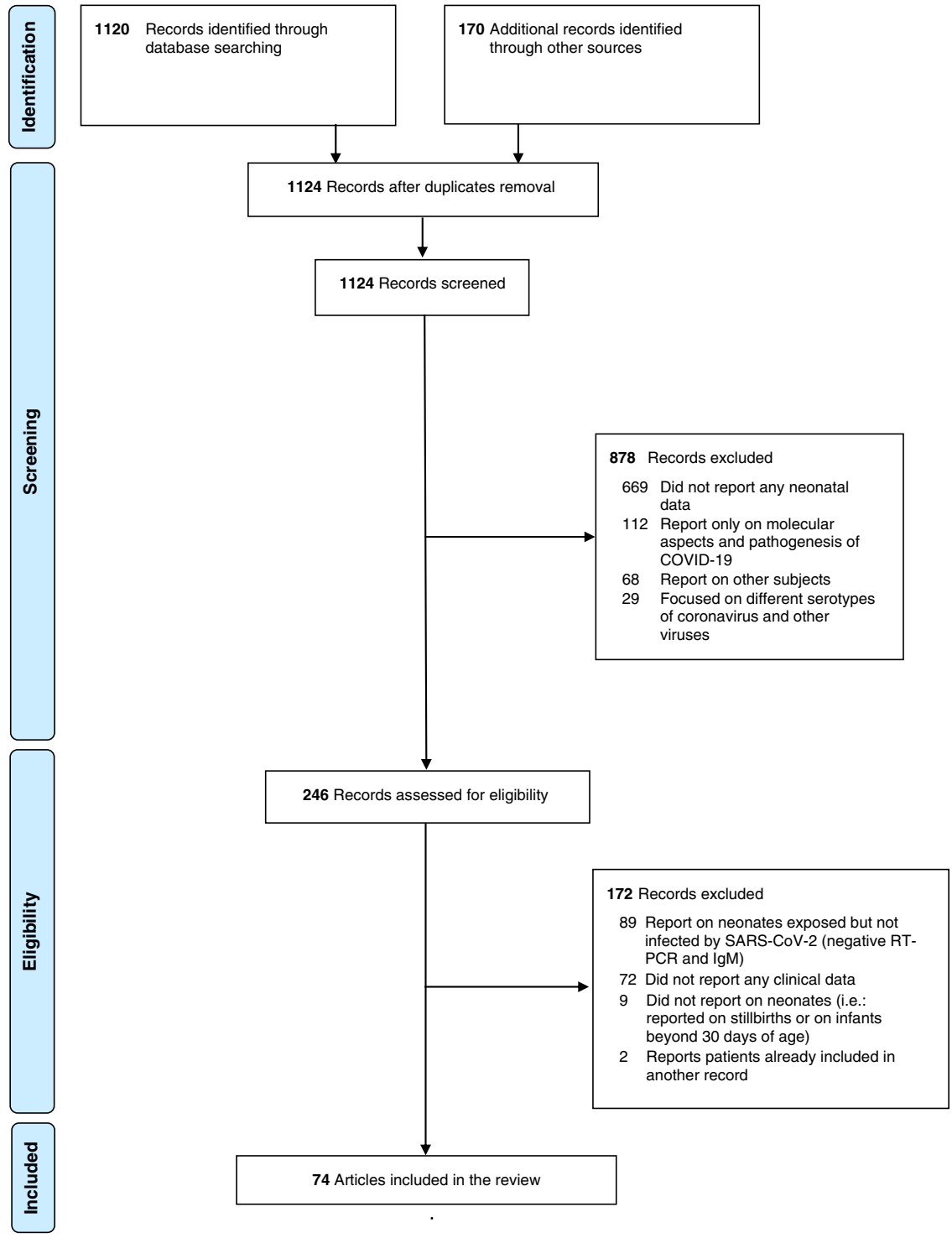

**Fig. 1 Study flow chart.** Flow chart prepared according to the PRISMA guidelines template for meta-analyses.

supportive care. Three out of 176 neonates (1.7%) died for reasons that seem unrelated to SARS-CoV-2 infection.

**Factors influencing occurrence of late neonatal infection**. Fig. 3 shows that the lack of mother-neonate separation from birth was significantly associated with the incidence of late (i.e. occurring after the first 72 h of life) SARS-CoV-2 neonatal infection (OR 4.94 (95% CI: 1.98–13.08), $p = 0.0002$; adjusted OR 6.6 (95% CI: 2.6–16), $p < 0.0001$), while breastfeeding was not (OR 0.35 (95% CI:

0.09–1.18), $p = 0.10$; adjusted OR 2.2 (95% CI: 0.7–6.5), $p = 0.148$); models goodness of fit was always satisfactory (Hosmer-Lemeshow test $p = 0.654$ and $p = 0.573$, for the two regressions, respectively).

**Discussion**
Several cases of neonates infected by SARS-CoV-2 are reported in the literature and here we have systematically analyzed and synthesized them. Our findings confirm that SARS-CoV-2 can infect neonates and that the majority of these infections occur

**Table 1 Characteristics of articles included in the systematic review.**

| First author/reference No. | Article type | Country | Publication stage | Quality score | Overall quality | No. neonates |
|---|---|---|---|---|---|---|
| Abasse[17] | Case report | France[a] | PR | 6 | Good | 1 |
| Aghdam[18] | Case report | Iran | PR | 4 | Intermediate | 1 |
| Alonso Diaz[19] | Case report | Spain | PR | 5 | Intermediate | 1 |
| Alzamora[20] | Case report | Peru | PR | 3 | Intermediate | 1 |
| Ayed[21] | Case series | Kuwait | Preprint | 1 | Low | 2 |
| Banerjee[22] | Case series | India | PR | 2 | Low | 6 |
| Barbero[23] | Retrospective cohort study | Spain | PR | n.a. | Intermediate | 1 |
| Buonsenso[24] | Case series | Italy | PR | 4 | Intermediate | 1 |
| Carosso[25] | Case report | Italy | PR | 4 | Intermediate | 1 |
| Chacon-Aguilar[26] | Case report | Spain | PR | 5 | Intermediate | 1 |
| Coronado Munoz[27] | Case report | USA | PR | 4 | Intermediate | 1 |
| Demirjian[28] | Case report | UK | PR | 4 | Intermediate | 1 |
| Dong[29] | Case report | China | PR | 6 | Good | 1 |
| Dumpa[30] | Case report | USA | PR | 6 | Good | 1 |
| Eghbalian[31] | Case report | Iran | PR | 4 | Intermediate | 2 |
| Feng[32] | Case series | China | PR | 3 | Intermediate | 4 |
| Fenizia[33] | Case series | Italy | PP | 5 | Intermediate | 2 |
| Ferrazzi[34] | Case series | Italy | PR | 1 | Low | 3 |
| Garazzino[35] | Case series | Italy | PR | 2 | Low | 15 |
| Gonzales Brabin[36] | Case report | Spain | PR | 3 | Intermediate | 1 |
| Gordon[37] | Case report | UK | PR | 2 | Low | 1 |
| Gregorio-Hernandez[38] | Case series | Spain | PR | 5 | Intermediate | 3 |
| Groß[39] | Case series | Germany | PR | 5 | Intermediate | 1 |
| Han[40] | Case report | South Korea | PR | 6 | Good | 1 |
| Hantoushzadeh[41] | Case series | Iran | PR | 6 | Good | 1 |
| Hasan[42] | Case series | Bangladesh | Preprint | 6 | Good | 7 |
| Hu[43] | Case series | China | PR | 6 | Good | 1 |
| Ibarra Rios[44] | Case report | Mexico | Preprint | 5 | Intermediate | 1 |
| Kanburoglu[45] | Case report | Turkey | PR | 1 | Low | 1 |
| Kayem[46] | Case series | France | PR | 1 | Low | 1 |
| Kirtsman[47] | Case report | Canada | PR | 6 | Good | 1 |
| Knight[48] | Retrospective cohort study | UK | PR | n.a. | n.a. | 12 |
| Kulkarni[49] | Case report | India | PR | 6 | Good | 1 |
| L'Huillier[50] | Case series | Switzerland | PR | 1 | Low | 1 |
| Lorenz[51] | Case report | Germany | PR | 6 | Good | 1 |
| Martinez-Peres[52] | Case series | Spain | PR | 1 | Low | 4 |
| Marzollo[53] | Case report | Italy | PR | 6 | Good | 1 |
| Meredith[54] | Case report | UK | PR | 1 | Low | 1 |
| Meslin[55] | Case series | France | PR | 6 | Good | 4 |
| Mithal[56] | Case series | USA | PR | 3 | Intermediate | 4 |
| Needleman[57] | Case report | USA | PR | 4 | Intermediate | 1 |
| Ng[58] | Case series | UK | PR | 6 | Good | 1 |
| Oncel[59] | Case series | Turkey | PR | 6 | Good | 4 |
| Paret[60] | Case series | USA | PR | 3 | Intermediate | 1 |
| Patanè[61] | Case series | Italy | PR | 6 | Good | 2 |
| Patil[62] | Case series | USA | PR | 4 | Intermediate | 3 |
| Pierce-Williams[63] | Case series | USA | PR | 5 | Intermediate | 1 |
| Piersigilli[64] | Case report | Belgium | PR | 6 | Good | 1 |
| Precit[65] | Case report | USA | PR | 5 | Intermediate | 1 |
| Salik[66] | Case report | USA | PR | 1 | Low | 1 |
| Salvatori[67] | Case series | Italy | PR | 6 | Good | 2 |
| Savasi[68] | Case series | Italy | PR | 1 | Low | 4 |
| Schwartz[69] | Case series | Iran | PR | 5 | Intermediate | 19 |
| Siddhi[70] | Case report | UK | PR | 3 | Intermediate | 1 |
| Sinelli[71] | Case report | Italy | PR | 6 | Good | 1 |
| Sisman[72] | Case report | USA | PR | 6 | Good | 1 |
| Sola[73] | Case series | Latin America[b] | PR | 5 | Intermediate | 6 |
| Sun D[74] | Case report | China | PR | 6 | Good | 1 |
| Sun[75] | Case series | China | PR | 6 | Good | 1 |
| Verma[76] | Case series | USA | PR | 2 | Low | 1 |
| Vivanti[77] | Case report | France | PR | 6 | Good | 1 |
| Wang[78] | Case report | China | PR | 6 | Good | 1 |
| Wang X[79] | Case report | China | PR | 6 | Good | 1 |
| Wardell[80] | Case series | USA | PR | 6 | Good | 4 |
| White[81] | Case series | USA | PR | 6 | Good | 3 |
| Wu[82] | Case series | China | PR | 5 | Intermediate | 3 |
| Xia[83] | Case series | China | PR | 4 | Intermediate | 3 |
| Xiao[84] | Cross-sectional study | China | PR | n.a. | Intermediate | 2 |
| Yu[85] | Case report | China | PR | 4 | Intermediate | 1 |
| Zamaniyan[86] | Case report | Iran | PR | 6 | Good | 1 |
| Zeng H[87] | Case series | China | PR | 2 | Low | 2 |
| Zeng L[88] | Case series | China | PR | 6 | Good | 3 |
| Zeng LK[89] | Case report | China | PR | 6 | Good | 1 |
| Zhang[90] | Case series | China | PR | 5 | Intermediate | 2 |
| Total | | | | | | 176 |

For articles reporting both infected and non-infected neonates, only those with proven infection (defined as at least one positive nasopharyngeal swab, and/or the presence of specific IgM) were included in the review. The methodological quality of case reports and case series was evaluated using the Mayo Evidence-Practice Center tool[103], specifically dedicated to the quality evaluation of this type of articles and summarized both as a 0–8 score and as overall qualitative evaluation. The tool was not applied to the retrospective cohort or cross-sectional studies included in the review.
n.a. not applicable, PR peer-reviewed.
[a]This case was observed in Mayotte Island, an overseas French department in in the Indian Ocean.
[b]This study was conducted in seven neonatal units affiliated to the network of the Latin American Neonatal Society (Sociedad Iberoamericana de Neonatologia (SIBENA); in detail, positive SARS-CoV-2 neonates included in this review were from Peru ($n = 3$), Dominican Republic ($n = 2$) and Honduras ($n = 1$).

postnatally, although vertical transmission may be possible in ~30% of cases. Neonatal SARS-CoV-2 become clinically evident in half of the patients as they developed features of COVID-19. The clinical appearance of neonatal COVID-19 seems similar to those reported in older patients, both in terms of symptoms and laboratory or imaging abnormalities, and the outcome is generally favorable. Neonates who were not transiently separated from their mothers seem to have a higher incidence of SARS-CoV-2 infections occurring after the first 72 h of life.

These findings are important as they formally describe neonatal SARS-CoV-2 infection and partially fill our aforementioned knowledge gap. There are some interesting points to be highlighted. First, transplacental transmission of SARS-CoV-2 is indeed possible and this is corroborated by a consistent background of laboratory findings, since angiotensin-converting enzyme receptors are expressed in placental tissues[91], with the expression reaching a peak at the end of gestation[92], and SARS-CoV-2 may invade the placenta[13,14] potentially causing miscarriage[93]. Second, neonatal COVID-19 manifestations seem similar to those observed in adults, while fever seems to be more frequent in COVID-19 than in common neonatal infections[94] and no cases of neonatal ARDS[95] were evident.

The choice between rooming-in or mother-infant separation is an important one and the synthesis of available cases shows that the avoidance of separation might be associated with a higher risk of late-onset neonatal SARS-CoV-2 infections. This is potentially important since neonatal SARS-CoV-2 infections are

### Table 2 Basic data of the reported neonates.

| Neonates (176) | Summary statistics | Min–max range |
|---|---|---|
| Gestational age (weeks) | 36.9 (3.4) | 26–41 |
| Birth weight (grams) | 2782 (799) | 900–4500 |
| Birth weight Z-score | 0 [−0.8;0.8] | −2.41-2.41 |
| Male sex | 63 (62.4%)[a] | – |
| Caesarean section | 66 (37.5%) | – |
| 5′ Apgar score | 9 [8.5;10] | 2–10 |
| Postnatal age at the diagnosis (days) | 5 [2;15] | 0–30 |
| Symptomatic neonates | 97 (55.1%) | – |

Data are expressed as mean (standard deviation) or median [interquartile range], min–max or number (%), as appropriate.
[a]Male sex percentage is referred to 101 neonates, as gender data were missing for the others, despite repeated requests to the authors of the articles.

### Table 3 Distribution of clinical features in the subgroup of neonates presenting with signs or symptoms compatible with COVID-19.

| Clinical features | Neonates (%) |
|---|---|
| Respiratory | 51 (52.5%) |
| Fever | 43 (44.3%) |
| Gastrointestinal | 35 (36%) |
| Neurological | 18 (18.6%) |
| Hemodynamic | 10 (10.3%) |
| Others | 9 (9.2%) |

Clinical features are listed in order of frequency; multiple features are possible in a patient; percentage is calculated for the group of symptomatic neonates (n = 97). More details in the text.

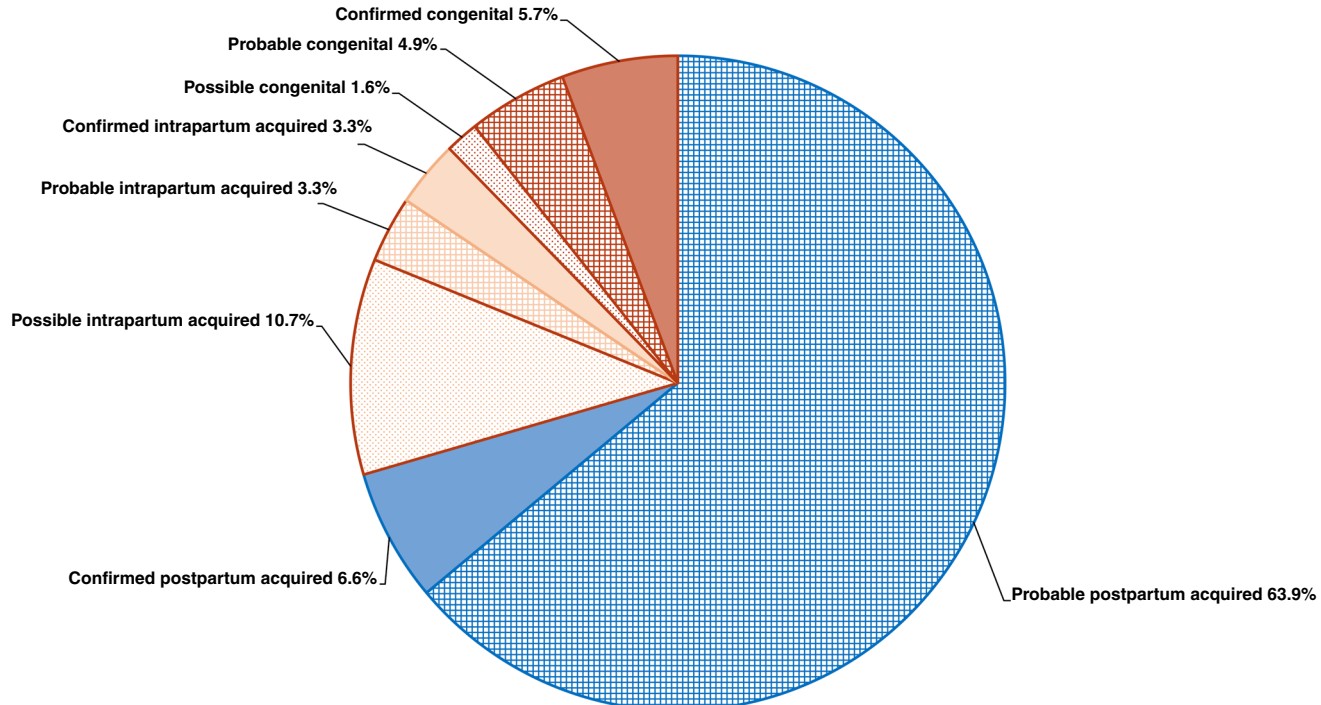

**Fig. 2 Classification of neonatal SARS-CoV-2 infections according to the definition of maternal, fetal, and neonatal SARS-CoV-2 infections.** Classification is based on a system including several virological tests (on placental tissues, amniotic fluid, cord and newborn blood or nasopharyngeal swabs), as well as the presence of clinical manifestations[16]. Cases are divided into: (1) congenital infections, (2) intrapartum acquired infections, or (3) postpartum acquired infections and into five mutually exclusive categories of the likelihood of infection: "confirmed", "probable", "possible", "unlikely", and "not infected". Classification was applied to 122 cases (for 54 neonates, data needed to classify the infection were missing despite repeated requests to the authors of the articles). Areas in blue depict the infections confirmed or supposed to be environmentally acquired (i.e.: postpartum), while areas in brown depict confirmed or supposed to be vertically (either intrapartum or congenitally) transmitted infections; numbers represent the %.

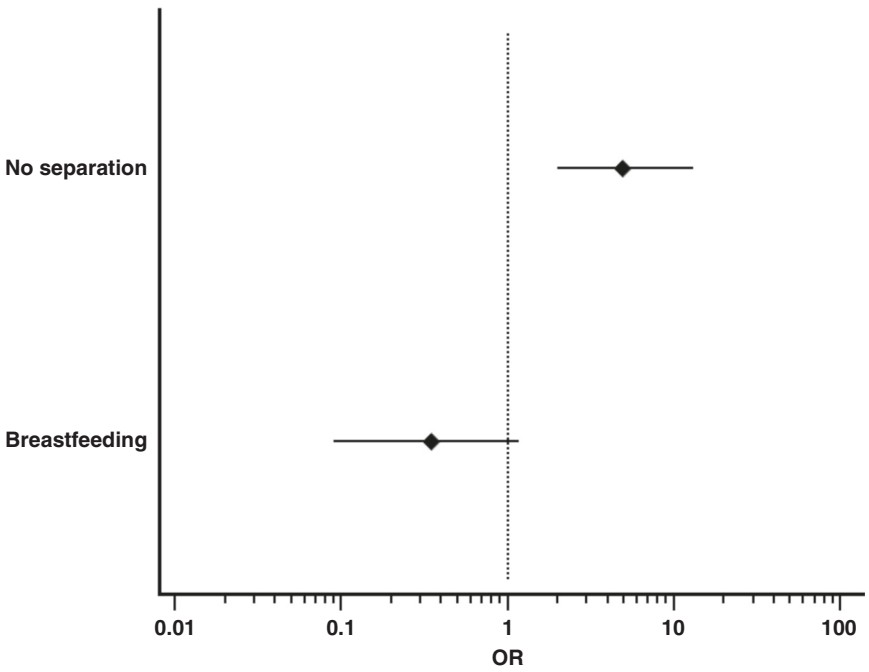

**Fig. 3 Effect of mother-neonate separation and breastfeeding on the occurrence of late SARS-CoV-2 infections.** Late infections are defined as those diagnosed after the first 72 h of life. Diamonds and horizontal lines represent the odds ratio (OR) and its 95% confidence interval (CI), respectively. Horizontal axis is on a log scale for better visualization; vertical hatched line represents OR = 1. Analysis was performed for 133 neonates for whom infection could have been classified as early- or late-onset. Figure illustrates OR 4.94 (95% CI: 1.98–13.08), $p = 0.0002$, for lack of mother-neonate separation from birth and OR 0.35 (95% CI: 0.09–1.18), $p = 0.10$ for breastfeeding. Analyses were performed with two-sided Fisher exact test.

more commonly acquired postnatally, through environmental exposure. There are wide differences in this matter between the clinical guidelines issued by scientific societies and authorities worldwide[96] (https://www.cdc.gov/coronavirus/2019-ncov/hcp/caring-for-newborns.html, https://www.rcog.org.uk/globalassets/documents/guidelines/2020-06-04-coronavirus-covid-19-infection-in-pregnancy.pdf) and these have been recently highlighted[97]. Thus, we believe that a correct and complete counselling should be given to families in order to allow a well-informed choice. This should factor in the benefits of mother-neonate bonding, the risk of neonatal infection, their usually (but not exclusively) benign outcome and the higher maternal contagiousness during the symptomatic period[98]. If rooming-in is chosen, appropriate hygiene advice and personal protective equipment should be given, as these may significantly reduce the risk of transmission[99]. Hygiene measures during mother-neonate contact were described inconsistently and only in a minority of articles[23,24,34,39,64,67,75], and, given the differences in guidelines worldwide[97], dedicated studies are warranted to understand what would be the ideal equipment and policy to be applied.

Breastfeeding does not seem to be associated with SARS-CoV-2 infections and this suggests that viral transmission through the milk, if any, should be rare. These findings seem to support the safety of expressed breast milk even when the mother is symptomatic and during mother-neonate separation. However, since few studies have investigated this matter and have yielded conflicting results[39,100], larger studies are needed to clarify this issue[15].

The results of this synthesis of neonatal SARS-CoV-2 infections are also consistent with a review performed in older children and adolescents from Asia. This review showed that SARS-CoV-2 infection seems mild in pediatric patients while presenting with the same clinical features or laboratory and imaging abnormalities[10]. Conversely, the analyzed neonates had a quite more frequent need for NICU hospitalization compared to

neonates of similar gestational age[101] and to older children[10]. However, not all these NICU-admitted babies may have actually needed critical care and this is not possible to be extrapolated with this type of analysis. The NICU-admission may be influenced by many factors such as local setting, logistics, and isolation policy, since the NICU could represent the only area to isolate infected babies and guidelines are extremely dissimilar between different countries and settings[97]. This should be avoided if NICU care is not actually needed, in order to avoid the shortage of NICU beds during the pandemic[102].

Since a significant proportion of infections are asymptomatic, it has so far been hard to ascertain the disease burden on neonates and the possibility of transmission to healthcare providers. We shed some light on this, although our findings may change as soon as the pandemic progresses and new experience is accumulated. Meanwhile, it is suggested that evidence derived from case reports and series is currently the best available and should be used to inform decision making until higher level of evidence is available[103].

This work has limitations. Although the quality of the reviewed reports is intermediate, we should remember that uncontrolled case descriptions and their synthesis are at the bottom of the evidence pyramid[103]. As this is a meta-analysis of mainly case series, the rating of the quality of evidence is 4, according to the Oxford Centre for Evidence-based Medicine classification (https://www.cebm.net/2009/06/oxford-centre-evidence-based-medicine-levels-evidence-march-2009/). However, the Grading of Recommendations, Assessment, Development, and Evaluation guidelines admit the decision-making process based on low-quality evidence in some particular circumstances[104] and the pandemic surely represents an extraordinary situation. The classification system for diagnosing maternal, fetal, and neonatal SARS-CoV-2 infections may be cumbersome, as it requires consideration of clinical data and of the results of several virological tests[16]: the relatively low number of confirmed infections may be

due to the difficulty in obtaining the virological tests in various samples (placenta, amniotic fluid, blood, swabs). However, this represents the best way so far to correctly identify these infections for epidemiological and clinical purposes and it should be promoted. Furthermore, the classification may be adjusted, and a case can be re-classified, if its likelihood of infection changes, as more information becomes available. Similarly, the availability of new diagnostic tests may lead to changes in the diagnostic criteria. Only one case was diagnosed based on positive IgM and negative RT-PCR[29], thus this is not likely to change the global results of our synthesis. We cannot exclude that some of the clinical features presented by the analyzed cases were due to other concomitant disorders, as this level of detail is not attainable with a synthesis of published reports; however, the analyzed data are coherent and consistent with those from older COVID-19 patients. We have almost no information on neonatal COVID-19 therapies, as these will require more experience and it is unclear if the neonatal SARS-CoV-2 infection may have long-term consequences, as no follow-up studies have been performed so far. Finally, we also enrolled manuscripts posted on preprint server while waiting for peer review and final publication. This might introduce a bias as the quality and reliability of these papers can be suboptimal. However, we provided our own peer review applying rigorous criteria and a specific quality evaluation tool to every and each article[105,103] and we actually have only three reports as preprint. Moreover, the role of preprints during pandemics has been widely recognized as they facilitate the spread of vital information and a recent analysis focused on COVID-19 found the quality of preprint similar to that of final publications[106].

In conclusion, the synthesis of uncontrolled cases of neonatal SARS-CoV-2 infection shows that infections mainly occur postnatally through environmental exposure, although nearly 30% of infections may be acquired vertically. Approximately half of the infected neonates develop clinically apparent COVID-19, which is often characterized by febrile status, features similar to those of older patients and favorable outcomes. Mother-neonate rooming-in is associated with a higher incidence of SARS-CoV-2 infections occurring after the first 72 h of life.

## Methods

**Protocol**. Prior to commencing the search, a detailed protocol was agreed to determine the databases to be searched, search modalities, eligibility criteria, and all methodological details. Several virtual meetings between the authors were organized and a dedicated online archive was created to share the data of the reviewed papers. Preferred Reporting Items for Systematic Reviews and Meta-Analyses (PRISMA) guidelines were followed throughout the whole project[107]. The French Ethical Committee for the Research in Obstetrics and Gynecology reviewed the work and confirmed that the institutional review board approval was unnecessary. The case studies were performed in agreement with principles of the Declaration of Helsinki and all analyzed data were anonymous and respecting local and European privacy regulations.

**Eligibility and exclusion criteria**. We looked for cohort, cross-sectional and case-control studies, as well as case series or case reports published as articles or letters to the editors describing neonates (i.e.: infants within the first 30 days of life) infected by SARS-CoV-2, as demonstrated by: (1) at least one positive real-time reverse transcription polymerase chain reaction (RT-PCR) on nasopharyngeal swabs, and/or (2) positive serology with detection of specific IgM. These laboratory tests had to be performed according to World Health Organization technical guidance principles or appropriate national guidelines[108]. Moreover, articles had to be published between 1 December 2019 and 30 August 2020. No language restriction was applied: non-English publications were translated personally by the investigators or using the Google translation service. We excluded conference abstracts, any report of neonates exposed to but not infected by SARS-CoV-2 (according to the aforementioned criteria), articles describing only stillbirths or children aged more than 30 days and paper reporting hypotheses or opinions without any clinical data. Duplicate reports and "grey" literature were also excluded.

**Information sources and search strategy**. We conducted an extensive search of the following databases: PubMed (http://www.ncbi.nlm.nih.gov/pubmed/), The Cochrane Library (https://www.cochranelibrary.com), Web of Science (https://clarivate.com/webofsciencegroup/solutions/web-of-science/), as well as BioXRiv (https://www.biorxiv.org) and MedXRiv (https://www.medrxiv.org) preprint archives. We used the following keywords or MeSH terms: "Coronavirus", "COVID-19", "SARS-CoV-2", "newborn", "preterm" and "neonates". We also hand-searched references cited in the eligible manuscripts or in review articles on the subject and the authors' personal archives. We used the following Boolean string: (COVID-19 AND neonates) OR (coronavirus AND newborn)) OR (coronavirus AND neonates) OR (coronavirus AND preterm) OR (SARS-CoV-2 AND newborn)) OR (SARS-CoV-2 AND preterm)) OR (SARS-CoV-2 AND neonates)) OR (COVID-19 AND preterm)) OR (COVID-19 AND newborn)) AND (("2019/12/01"[Date - Create]: "2020/08/30"[Date - Create])).

**Study selection**. Articles were assessed by two independent researchers (RR and AJV), as recommended by the quality standards issued by the Meta-analysis Of Observational Studies in Epidemiology (MOOSE) Group[109]. Investigators evaluated abstracts and (where necessary) the full text of each article, excluding those not meeting the eligibility criteria and removing duplicates. The CARE (Consensus-based Clinical Case Reporting Guideline Development) recommendations, specifically dedicated to case reports and series, were followed during the evaluation process[105]. If an article was eligible but reported data on neonates mixed with those of older children (beyond the first month of life), neonatal data were directly extracted or the authors were contacted to provide the needed data. Authors were also contacted when more information was needed and at least two emails were sent to the corresponding author, 1 week apart, asking for additional data. If discrepancies or uncertainties persisted, they were resolved by discussion between the two independent researchers and, if no agreement was reached, with a third researcher (DDL). All articles finally deemed eligible were collected using Zotero (5.0.87, Roy Rosenzweig Centre for History and New Media, Fairfax, VI – USA) and used for the review.

**Data collection process**. We developed a dedicated online data extraction sheet (Excel 16; Microsoft Corporation, Redmond, WA-USA), pilot-tested it on three randomly selected manuscripts, and refined it accordingly. Data from included records were extracted independently by two investigators (RR and AJV) using the data extraction sheet and then cross-verified. If some data were lacking, at least two emails were sent to the corresponding authors one week apart to ask them for the missing data. If discrepancies or uncertainties persisted, they were resolved by discussion between the two independent researchers and, if no agreement was reached, with a third researcher (DDL).

**Data items**. Data collected included article type and details, number of enrolled patients, and their basic characteristics, as mean gestational age, birth weight, sex, mode of delivery, Apgar score, postnatal age at the diagnosis and comorbidities. We also recorded all clinical signs and symptoms, imaging and laboratory findings, use of antiviral therapies, isolation and feeding policies, need for neonatal intensive care unit (NICU) admission and mortality. According to postnatal age at the diagnosis, the SARS-CoV-2 infection was classified as early- or late-onset (i.e. occurring ≤72 or >72 h of life, respectively), as commonly done for neonatal sepsis. We recorded any RT-PCR result in placental tissue, amniotic fluid, cord or newborn blood, urine, stool, and swabs. We also recorded the presence of SARS-CoV-2-IgM, and classified the cases according to the classification system defining maternal, fetal and neonatal SARS-CoV-2 infections[16]. In detail, cases were divided into: (1) congenital infections, (2) intrapartum acquired infections, or (3) postpartum acquired (i.e. environmentally acquired) infections and into five mutually exclusive categories of the likelihood of infection: (a) confirmed (strong evidence of infection with confirmatory microbiology), (b) probable (strong evidence of infection but confirmatory microbiology lacking), (c) possible (evidence suggestive of infection but incomplete), (d) unlikely (little support for diagnosis but infection cannot be ruled out), and (e) not infected (no evidence of infection)[16]. Criteria defining the likelihood of infection were published earlier and are available as Supplementary Material[16].

We considered outcome data, respectively for our primary and secondary objective: (1) the identified transmission route and all clinical features[16], and (2) the occurrence of late neonatal infection (that is, those diagnosed after the first 72 h and within the first 30 days of life).

**Assessment of risk of bias**. Since we expected the majority of analyzed articles to be case reports or case series, we decided to evaluate their methodological quality according to four domains (selection, ascertainment, causality, and reporting). To perform this evaluation, we used the Mayo Evidence-Based Practice Centre tool, which is specifically dedicated to the evaluation of case report/series quality[103]. Two investigators (RR and AJV) independently summarized the results of this evaluation by aggregating the eight binary responses into a 0–8 score. Evaluation results were also qualitatively summarized, as recommended by the tool creators[103]. If discrepancies or uncertainties persisted, they were resolved by discussion

between two researchers (RR and AJV) and, if no agreement was reached, with a third researcher (DDL).

**Summary measures and synthesis of results**. Cumulative estimates of event rates (frequency) were reported as a percentage for the need of neonatal resuscitation or NICU admission, transmission route, isolation and feeding policies, and for each clinical manifestation, imaging and laboratory findings or therapy. Continuous data were described as mean (standard deviation) or median [interquartile range], as appropriate; minimum and maximum values were also reported: this was the case of basic clinical data (such as birth weight or postnatal age at the diagnosis) and length of NICU stay.

The effect of mother-neonate separation and breastfeeding on the incidence of late-onset neonatal infections was studied with two-sided Fisher's exact test and association between variables was expressed by means of the odds ratio (OR) and 95% confidence interval (CI). Calculations and statistics were performed with Excel 16 (Microsoft Corporation, Redmond, WA-USA) and MedCalc 13.3 (MedCalc, Ostend, Belgium). $p$-values < 0.05 were considered statistically significant.

**Additional analyses**. The effect of mother-neonate separation and breastfeeding on the incidence of late-onset neonatal infections was also investigated using multivariate logistic regressions, with enter method, adjusting for the quality of reviewed article (evaluated by the specific tool[103] see above). Results were expressed as adjusted OR and 95% CI; model goodness-of-fit was evaluated with Hosmer-Lemeshow test.

**Reporting summary**. Further information on research design is available in the Nature Research Reporting Summary linked to this article.

## Data availability

All summary data generated during this study are included in this published article. Raw data used for the analyses are available upon request or presented in the original reviewed articles which have been retrieved from the following publicly available databases: PubMed (http://www.ncbi.nlm.nih.gov/pubmed/), The Cochrane Library (https://www.cochranelibrary.com), Web of Science (https://clarivate.com/webofsciencegroup/solutions/web-of-science/), BioXRiv (https://www.biorxiv.org), and MedXRiv (https://www.medrxiv.org).

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

## Acknowledgements

We are grateful to the authors of reviewed papers who provided details needed for the synthesis. We also thank the "Emergencia Obstetrica España" Group for the kind collaboration.

## Author contributions

Acquisition, analysis, or interpretation of data: R.R., A.V., A.B., C.V.F., and B.L.; drafting of the paper: R.R.; calculations and statistical analysis: A.V. and D.D.L.; administrative, technical, or material support: A.B., C.V.F., and B.L.; study concept, design, and supervision: D.D.L. Critical revision of the paper for important intellectual content: all authors. R.R. and A.V. contributed equally and should be considered co-first authors. All authors take full responsibility for the content of the paper.

## Competing interests

The authors declare no competing interests.
