## [Peer Review File · Nature Communications]

REVIEWER COMMENTS

Reviewer #1 (Remarks to the Author):

General comments

The manuscript of Raschetti R and Colleagues entitled "Neonatal SARS-CoV-2 infections: systematic review, synthesis and meta-analysis of reported cases" is aimed at describing the findings of a systematic review and meta-analysis focused on neonatal infections caused by SARS-CoV-2.

The topic is potentially interesting and the manuscript is well-written, although not novel. Some methodological shortcomings can be found, from the unclear identification of the selection criteria to the issues related to some electronic search engines, the analysis plan.

Detailed comments

Introduction

- They should explain the acronyms SARS-CoV-2 and COVID-19.
- They should detail how many positive cases, how many cases of disease in the epidemiological background.
- All the lines should be supported by one or more scientifically sound references.
- They should include dermatological symptoms and clinical signs.
- They should better define the role played by the risk factors in the occurrence of the disease.
- They should define the prevalence of asymptomatic infection in children and adults.
- The aim is unclear: they should better define the primary objective and two/three secondary objectives.

Methods

- The serological criterion for the infection is disputable based on the reliability of the tests and on the gaps of knowledge in the immunopathogenesis.
- Case-reports or letters to the editors do not provide sufficient data to address the issues
- Why did they select the date of the 1st December 2019?
- Exclusion criteria should be defined.
- Preprint archives could be a biased source of information: they did not include peer-reviewed articles and, then, the reliability is poor.
- Explanation of the categories of the likelihood of infection should be provided.

- How did they define the timing of the infection?
- Which outcomes did they record?
- To assess the quality of the case-reports they could use an ad hoc tool published in the BMJ.
- The statistical plan is inappropriate. It is generic and not tailored on what they want to prove.

Results

- References should be used for every finding.
- Characteristics of the studies should be summarized.
- When they describe the results, proportions or summary estimates for continuous variables should be adopted (e.g., symptoms).

Reviewer #2 (Remarks to the Author):

This manuscript entitled "Neonatal SARS-CoV-2 infections: systematic review, synthesis and meta-analysis of reported cases" by Dr De Luca et al is a descriptive meta-analysis of the published case series and case reports of neonates with COVID-19. They identified 117 infected newborns.

Overall, it is a well-written study that tries to describe clinical presentation and the risks factors to acquire the infection in the neonatal period

Comments:

In the Result section

Page 7 and Table 2

It would be helpful if the authors could provide additional information, if available, regarding the 36 newborns admitted to the NICU in gestational age sub-categories, e.g., moderate preterm (32 to 33 weeks) and late preterm (34 to 36 weeks). Similarly, it would be helpful if the authors could provide data on the number of neonates in birthweight sub-categories, e.g., normal ($\geq 2500\text{g}$), low (1500-

2499g), very low (1000-1499g), and extremely low (<1000g) and the admission diagnoses and if COVID-19 related or due to prematurity or other causes

Page 8, paragraph 2

It would be important to understand if correct infection control precautions were undertaken by mothers (mask, hand hygiene, etc) when near the newborn or infection occurred mainly because of lack of the above practices.

In the Discussion section

Page 9

Even though from these reported cases there is a high incidence of post-natal transmission, there is no description if precautions were undertaken. There is now new published data regarding the safety of rooming-in and against mother-newborn dyad separation if precautions are undertaken. The new American Academy of Pediatrics Guidelines and a recent study published online July 23, 2020 [https://doi.org/10.1016/S2352-4642\(20\)30235-2](https://doi.org/10.1016/S2352-4642(20)30235-2) in Lancet Child and Adolescent Health are supporting not to separate the mother and newborns. I would revise this section of the discussion according to these new findings and recommendations and add the 2 references. As it is now, the discussion seems favoring the separation of the dyad and could give the wrong message

Reviewer #3 (Remarks to the Author):

I agree with the first reviewer regarding the assessment of study quality. The authors have ignored major components of the systematic review. The systematic reviews aim to assess the quality of included articles to disclose the risk of bias and conclude the level of evidence. The concluded level of evidence in systematic reviews is an important source for both future research and clinical recommendations. Therefore, I suggest assessing the study quality and report the results inside the text. The statistical analysis is performed well, but the authors did not include the Forest plot. I highly recommended the authors to include the forest plot.

REVIEWER COMMENTS

Reviewer #1

The manuscript of Raschetti R and Colleagues entitled "Neonatal SARS-CoV-2 infections: systematic review, synthesis and meta-analysis of reported cases" is aimed at describing the findings of a systematic review and meta-analysis focused on neonatal infections caused by SARS-CoV-2. The topic is potentially interesting and the manuscript is well-written, although not novel. Some methodological shortcomings can be found, from the unclear identification of the selection criteria to the issues related to some electronic search engines, the analysis plan.

We thank the Reviewer for the thoughtful comments! These gave us the possibility to improve the manuscript in several points. We followed all of them and provided requested modifications or details into the text; we also added relevant references and, in some cases, provided additional analyses to substantiate our findings and answers. Moreover, we provided detailed explanations where we felt there was a misunderstanding.

Please see below our point-to-point reply; all changes are in red font through the manuscript.

We would highlight that some of the required issues were already addressed in the first manuscript version and we believe that they were overlooked by misunderstanding and/or because we were not enough clear (for example, the tool to evaluate the quality of case reports was already in methods (reference 13, now updated to reference 22), results and tables; search details were already described in methods, studies characteristics were already described in Tab.1, PRISMA checklist (reference 10, now updated to reference 18), was strictly followed etc...). **In these cases, we anyway rephrased the text to be clearer.** We are willing to provide further modifications if required. Thanks again!

Introduction

C2.-They should explain the acronyms SARS-CoV-2 and COVID-19. **A2. Done**

C3. -They should detail how many positive cases, how many cases of disease in the epidemiological background. **A3.** This is now better detailed with the most updated figures from the last WHO situation report, from the experience of the Italian Village Vo' (published in *Nature*, reference 3) and of the Diamond Princess boat (published in *NEJM*, reference 4).

C4.-All the lines should be supported by one or more scientifically sound references. **A4.** We especially thank the Reviewer for this comment. Originally, we did not insert a citation for every single important assumption in the intro for brevity. However, the Reviewer is totally right and now we have inserted several references, paying attention to slightly rephrase according to the very last knowledge on COVID-19 physiopathology and giving preferences to large manuscripts and meta-analysis published in *Nature* or other major journals.

C5.-They should include dermatological symptoms and clinical signs. **A5. Done**

C6.-They should better define the role played by the risk factors in the occurrence of the disease.

A6. Added with references. Please see also point C4.

C7.-They should define the prevalence of asymptomatic infection in children and adults.

A7. We thank again the Reviewer as this gives us the possibility to update the intro with the most recent and high-quality data on the subject. We have now rephrased this part highlighting that, according to recent studies in *NEJM* and *Lancet* the incidence of asymptomatic infections seems lower in children (16-22%, references 12 and 13) than in adults (40-45% as we reported above, see references 3-4). However, these data must be considered preliminary as the knowledge on pediatric SARS-CoV-2 infection is still limited and we lack of data coming from longitudinal large cohorts (as our EPICENTRE that is still ongoing – see reference 10). These are available only in adults (and have been meta-analyzed in reference 4). As such, the limited pediatric knowledge contributed to the uncertainty about neonatal infections as well, and represents the background for our work. Thanks to the Reviewer, we hope that this rephrased part is now guiding better the reader directly towards our study aims.

C8. The aim is unclear: they should better define the primary objective and two/three secondary objectives.

A8. Thanks to the Reviewer's suggestions and consistently with point C7, this part has been rephrased and is now more understandable. We had two objectives: the primary was to describe the clinical characteristics and route of transmission of neonatal SARS-CoV-2 infections (this has never been done, since the beginning of pandemics). Our secondary objective was to clarify the effect of mother-neonate separation and breastfeeding on the incidence of neonatal infections acquires postnatally. Added in the Intro, pag.4, 1st par.

Methods

C9. -The serological criterion for the infection is disputable based on the reliability of the tests and on the gaps of knowledge in the immunopathogenesis.

A9. We agree in principle with the Reviewer: in fact, the uncertainty on neonatal COVID-19 constitutes the background for our work (see also point C7). However, for the purpose of our analysis, this point has no relevance since only 1 neonate was diagnosed by having positive IgM only and negative PCR on nasopharyngeal swabs (*Dong et al, JAMA 2020*, reference 32). According to the classification of neonatal and perinatal SARS-CoV-2 infections, this case fully qualifies as “possible congenital infection in a live born neonate” as there was “No detection of the virus by PCR in nasopharyngeal swab at birth (collected after cleaning baby) BUT presence of anti-SARS-CoV-2 IgM antibodies in umbilical cord blood or neonatal blood collected within first 12 hours of birth or placental tissue” (classification in reference 17, pag.566)

All the other cases were diagnosed as infected based on at least one positive PCR.

Thus, following Reviewer’s suggestion, we specifically described this case with this peculiarity in results, pag.8, 3rd par and also recalled this in discussion, study limitations, pag.12. We are willing to totally exclude the unique patient with positive IgM if required by the Reviewer or the Editor.

C10.-Case-reports or letters to the editors do not provide sufficient data to address the issues.

A10. We respectfully disagree and we believe that there is a misunderstanding here. While the vast majority of meta-analyses are based on RCTs or large observational studies, these studies cannot be used for the purpose of our work for several reasons:

1. There is no RCT and no large observational study focused on neonatal SARS-CoV-2 infection. This is because pandemics is a quickly evolving situation and because COVID-19 is a disease affecting much more older adults than children (thus there are few cases in newborn infants). As such, all our knowledge about neonatal SARS-CoV-2 infections and COVID-19 comes from case reports and series.
2. Case reports and series, if well prepared, do provide all the relevant informations to address our objectives (that is, primarily to describe clinical characteristics and route of transmission of neonatal SARS-CoV-2 infections and, secondarily, to clarify the effect of mother-neonate separation and breastfeeding on the incidence of late, environmentally-acquired neonatal infections). In fact, similar works have been already performed with similar primary aims in older children (*Castagnoli et al, JAMA Pediatr 2020* – reference 11), in pregnant women (*Khalil A et al, EClinMedicine 2020*), in obese patients (*de Carvalho Sales-Peres S et al, Obes Res Clin Pract 2020*) and in other particular subset of patients representing rare situations.
3. Meta-analyses of RCT or observational studies are useful to clarify the impact of a given intervention, compared to a control, on the outcome in a given population (PICO questions methodology). This is an *objective completely different* from ours, as there is no intervention possible for infected neonates, because we are in a much earlier knowledge phase and we need to understand their clinical characteristics and transmission route first (please see our study purpose, point C8 and also point C16).

Therefore, the synthesis/meta-analysis of case reports and series, although less common than the meta-analysis of RCTs and observational studies, is a particularly interesting technique for a relatively rare condition lacking of other studies (such as neonatal SARS-CoV-2 infection).

An example of the same situation may be found in the recently published meta-analysis of case reports on the epidemiology and clinical manifestations of mucormycosis (Jeong W et al, Clin Microbiol Infect 2019 - doi.org/10.1016/j.cmi.2018.07.011).

Moreover, during an emergency situation, where research resources are mainly dedicated to other issues and large neonatal observational studies are lacking, the meta-analysis of case reports really represents the best study design, because it allows to convey important messages to clinicians on the frontline without waiting for RCT and large observational studies. It is however, important to evaluate case report/series with the best methodology, which is what we did using a dedicated scoring system (Mayo Evidence-Based Practice Centre - Murad MH et al, *BMJ Evid Based Med 2018* – reference 17) and CARE guidelines (reference 21). In order to improve the clarity and limitations of our work, we are willing to add some of these considerations if required by the Editor or the Reviewer.

C11. -Why did they select the date of the 1st December 2019? **A11.** This date was just chosen as a safety threshold because the first cases of COVID-19 were observed in China during December 2019 and officially declared to the local WHO office on dec 31 (see WHO notice: <https://www.who.int/csr/don/05-january-2020->

pneumonia-of-unknown-cause-china/en/). Thus, we decided to start our search on Dec 1st, to be sure not to miss any case.

C12. -Exclusion criteria should be defined. **A12.** We are extremely grateful to the Reviewer as this was a forgotten point! We thought to have described these implicitly in Fig.1, but we have now clearly added all the exclusion criteria, in methods, pag. 5, 1st par.

C13. -Preprint archives could be a biased source of information: they did not include peer-reviewed articles and, then, the reliability is poor.

A13. We fully understand the Reviewer's point of view and we are grateful as this is a relevant issue! This was considered important by the Editor as well and we want to immediately say that **we did some additional analyses and added their results and related comments to the text following Reviewer's point of view (see below).**

However, we believe that the role of preprint has rapidly changed during the pandemics. Please consider the following:

1. Preprints facilitate the spread of potentially important information that may be extremely useful during a rapidly evolving situation such as a pandemic new disease for which clinicians are unarmed and lacking knowledge. The use of preprint has exploded in recent times and this has been acknowledged by the general media (see for instance: <https://cen.acs.org/policy/publishing/Pandemic-puts-preprints-first/98/i22>) and by academic forums (see: <https://rapidreviewscovid19.mitpress.mit.edu/pub/k8h2xox0/release/1>).
2. This has also been recognized by a specific analysis of preprints on COVID19 epidemiology, recently published in *Lancet Global Health*, showing that the quality of preprints was not far from that of final publications and concluding: "*Our work showcases the powerful role preprints can have during public health crises because of the timeliness with which they can disseminate new information. [...] Without question, primacy and peer-reviewed publications are key metrics in individual professional advancement (eg, academic promotion); nevertheless, the impact of preprints on discourse and decision making pertaining to the ongoing COVID-19 outbreak suggests that we must rethink how we reward and recognise community contributions during present and future public health crises*" (Majumder MS et al, *Lancet Global Health* 2020 - doi.org/10.1016/S2214-109X(20)30113-3 – reference 97).
3. *Nature Communications*, as all Nature family journals, has its own preprint server and authors are encouraged to post submitted articles there (see: <https://www.nature.com/nature-research/editorial-policies/preprints-and-conference-proceedings>). We did so for the present work and a previous one recently published. Obviously, the existence of a preprint server coupled with a high impact journal is a further guarantee of quality.
4. Beside Nature family journals, other major scientific publishers facilitate journals editors and preprint archives to work together in order to "*complement traditional journal publishing, adding speed, openness, and faster feedback for researchers*" (see for instance: John Wiley and Sons Publishing: <https://www.wiley.com/network/archive/preprints-publishing-and-a-pandemic-your-questions-answered>).
5. According to the 2016 Statement on data sharing in public health emergencies, (<https://wellcome.ac.uk/what-we-do/our-work/statement-data-sharing-public-health-emergencies>) many healthcare systems, publishers and other bodies agreed to share still unpublished, under review data, and currently all the major journals are sending manuscripts on COVID-19 to the WHO in form of preprint in order to diffuse the knowledge as quicker as possible.
6. For the sake of our work, we should consider that we provided the first intention peer review for cases series/reports available only as preprint at the time of writing. In fact, they have been independently evaluated by two investigators (see methods, study selection and data collection process, pag.5-6) using a specifically dedicated scoring system (Mayo Evidence-Based Practice Centre - Murad MH et al, *BMJ Evid Based Med* 2018 – reference 22) and following the CARE guidelines (reference 21). The same tool and criteria have also been used for the already published articles (thus all articles have been evaluated in the same way).

Anyway, this is no more a very relevant issue because, since our first submission, many preprint articles have been already peer reviewed and finally published (we have updated the reference list); only two articles remain on preprint archives.

Having said that, we believe that this is an important issue and we acknowledged it as study limitation (Discussion, study limitations, end of pag.12) and cited the preprint analysis published in

Lancet Global Health 2020 (reference 97). We also explicitly declared which manuscript was a preprint and which one has been peer reviewed and published (Tab.1).

Finally, to be extra sure about the quality of our results, in relation to the quality of reviewed articles, we performed multivariate logistic regressions, with enter method, adjusting for the quality of reviewed article (evaluated by the specific tool (Mayo Evidence-Based Practice Centre - Murad MH et al, *BMJ Evid Based Med* 2018 – reference 22). The regressions had the incidence of late onset neonatal infections as dependent variable, and the mother-neonate separation or breastfeeding as independent variables. Adjusted OR were consistent with crude OR and our results did not change. This is now described in Methods, additional analyses, pag.8, 1st par and at the end of results, end of pag.9.

C14. -Explanation of the categories of the likelihood of infection should be provided. A14. These criteria are published elsewhere (reference 17) but we have now better described this in Methods, data items, pag.6-7 and the detailed criteria for each likelihood of infection category are listed in a supplementary material. This allows the reader to understand the transmission route without the need to retrieve the original publication describing the classification system (see also point C15 below).

-C15. How did they define the timing of the infection? A15. The timing of infection has been defined using the strict criteria provided by the Classification system and case definition for SARS-CoV-2 infections in fetuses and neonates published in Shah, PS, et al. *Acta Obstet Gynecol Scand* 2020 (reference 17).

Therefore, the timing was considered as congenital (intrauterum transmission before the delivery), intrapartum (transmission at the delivery) or postpartum (after at least 48h from birth); all infections had to be apparent within the first 30 days of life (neonatal period) as detailed in methods, eligibility and exclusion criteria, end of pag.4). All these criteria are now described in details in the supplementary material, reporting the original criteria listed from reference 17. This allows the reader to understand the transmission route and timing without the need to retrieve the original publication describing the classification system (see also point C14 above).

C16.-Which outcomes did they record? A16. There is a misunderstanding here. As also explained above (please see point C10), we did a synthesis and meta-analysis of case reports and series. This is very different from the meta-analysis of RCTs or observational studies. RCTs and observational studies report an “intervention” in a given patients’ population (following randomization or not) and the effect of that intervention is evaluated by a given outcome, which is measured both in treated patients and in comparators (i.e.: in an untreated, or differently treated, patients’ population). This is resumed in the so-called PICO (patient-intervention-comparator-outcome) questions acronym.

A meta-analysis of case reports/series is different because these studies usually do not have an intervention and, by definition, cannot have an external comparator. Moreover, in our case, we could not have an intervention since neonatal SARS-CoV-2 infection does not have any specifically validated or even proposed therapy.

In fact, all meta-analyzed case reports/series did not focus on a therapy but rather on the clinical characteristics and route of transmission of neonatal infections, whose description represents our primary objective (please see point C8). **Thus, we did not analyze an intervention and could not have an intervention, nor an outcome potentially modified by that intervention. An example of the same situation may be found in the recently published meta-analysis of case reports on the epidemiology and clinical manifestations of mucormycosis (Jeong W et al, *Clin Microbiol Infect* 2019 - doi.org/10.1016/j.cmi.2018.07.011).**

However, we wanted to increase readability of our manuscript, especially because the message should be understandable to any reader during this worldwide emergency situation. Therefore, following the Reviewer’s suggestion we better specified our outcome, which correspond to our study objectives. This is now better specified in Methods, data items, pag.7, 1st par.

C17.-To assess the quality of the case-reports they could use an ad hoc tool published in the BMJ.

A17. We already did so. We used the Mayo Evidence-Based Practice Centre tool, which is the specific tool dedicated to the evaluation of case report/series quality (published in Murad MH et al, *BMJ Evid Based Med* 2018 – reference 22). This is exactly the same tool suggested by the Editor. This tool has been already used in similar meta-analyses of case reports/series (for ex.: Bazerbachi F, et al. *Gastroenterol Rep* 2017).

This was already described in Methods, Assessment of risk of bias, pag.7, 3rd par. Results of this evaluation were already reported in Tab.1 and Results, pag.8, 2nd par. Maybe this has been overlooked, so we slightly rephrased the methods to highlight the use of this tool.

C18.-The statistical plan is inappropriate. It is generic and not tailored on what they want to prove.

A18. We thank the Reviewer to push us improving this. This section was indeed very generic. We have now rephrased and better detailed how we described and treated every variable, according to our objectives (please also see point C8 and C16). We believe that this is much clearer now but we are willing to provide further amendments if needed.

Moreover, we added, as additional analysis, two multivariate logistic regressions, adjusting for the quality of reviewed article (evaluated by the specific tool (Mayo Evidence-Based Practice Centre - Murad MH et al, *BMJ Evid Based Med* 2018 – reference 22). The regressions had the incidence of late onset neonatal infections as dependent variable, and the mother-neonate separation or breastfeeding as independent variables. Adjusted OR were consistent with crude OR and our results did not change. This is now described in Methods, additional analyses, pag.8, 1st par and at the end of results, end of pag.9 (see also point C13).

Results

C19. -References should be used for every finding. **A19.** Following Reviewer's suggestion, we added all the references for each study while describing the frequency of each clinical manifestation (Results, pag.8-9)

C20. -Characteristics of the studies should be summarized. **A20. Characteristics of the studies were already summarized in Tab.1. Furthermore, we have added to this table a new column to indicate if the article was a preprint or an already peer-reviewed manuscript. This table illustrates all the basic informations for each article and it is hard to add other details for two reasons. First, because there are a lot of case reports and the Table (already big) would become extremely difficult to print and read. Second and more important, because all the other characteristics are patients' data that constitute our results and are already summarized in results section.**

This is a main difference between our technique and the more common meta-analysis of RCTs where results consist of the effect of an intervention (please see also point C10). Thanks to the Reviewer (point C19 above), we also added the reference for each article while describing results and this makes findings more directly readable. We remain available to discuss new forms of data presentation if needed.

C21.-When they describe the results, proportions or summary estimates for continuous variables should be adopted (e.g., symptoms). **A21. All this was already done and available in results, pag.8-9, in Fig.2 and in Tab.3.** We are happy to modify the data presentation if required by the Editor or Reviewer.

Reviewer #2

This manuscript entitled "Neonatal SARS-CoV-2 infections: systematic review, synthesis and meta-analysis of reported cases" by Dr De Luca et al is a descriptive meta-analysis of the published case series and case reports of neonates with COVID-19. They identified 117 infected newborns. Overall, it is a well-written study that tries to describe clinical presentation and the risks factors to acquire the infection in the neonatal period. We thank the Reviewer for the thoughtful comments! These gave us the possibility to improve the manuscript in some points. We followed all of them and provided requested modifications or details into the text and also added suggested references. Please see below our point-to-point reply; all changes are in red font through the manuscript. Thanks again!

C22. In the Result section

It would be helpful if the authors could provide additional information, if available, regarding the 36 newborns admitted to the NICU in gestational age sub-categories, e.g., moderate preterm (32 to 33 weeks) and late preterm (34 to 36 weeks). Similarly, it would be helpful if the authors could provide data on the number of

neonates in birthweight sub-categories, e.g., normal ($\geq 2500\text{g}$), low (1500-2499g), very low (1000-1499g), and extremely low ($< 1000\text{g}$) and the admission diagnoses and if COVID-19 related or due to prematurity or other causes.

A22. We thank the Reviewer for this interesting comment. Of the NICU-admitted neonates, two were extremely preterm (of 26 and 27 weeks' gestation; birth weight 960 and 1410g, respectively), three were preterm (31 and 32 weeks' gestation; birth weight 1580, 1620 and 2100g), two were moderately preterm (both 33 weeks' gestation; birth weight 2350 and 2970g) and five were late preterm (of 34, 35 and 36 weeks' gestation; birth weight 2540, 2686 and 2930 g; birth weight was not recorded for the remaining); all remaining NICU-admitted neonates were term infants.

We added these data in Results, end of pag.8.

Of note, we already wrote in the discussion that unnecessary NICU admission should be avoided. Consistently with these data, we slightly expanded this part and acknowledged that it is not possible to judge if babies were admitted to the NICU for the actual need of critical care or just for isolation, as this level of detail is not reachable with our study design. In some settings, the NICU may be the only place to isolate infants and guidelines are extremely variables on this point.

We expanded this in Discussion, pag.11, 3rd par. and added as reference our review on the diversity of clinical management guidelines (Yeo KT, Oei JL, De Luca D, et al. *Review of guidelines and recommendations from 17 countries highlights the challenges that clinicians face caring for neonates born to mothers with COVID-19* [published online ahead of print, 2020 Jul 27]. *Acta Paediatr.* 2020;10.1111/apa.15495) as reference 89.

C23. It would be important to understand if correct infection control precautions were undertaken by mothers (mask, hand hygiene, etc) when near the newborn or infection occurred mainly because of lack of the above practices. **A23.** We thank the Reviewer for this important comment, as this is a hot issue. We double checked all the articles and contacted some authors and unfortunately, this information has not been reliably recorded or is unavailable. Only six articles (references 27,36,41,60,63 and 68) reported the use of masks in proximity or while caring for the neonate. However, the type of mask and PPE used is not consistent and/or not always specified. As we know (see also point above C22 and our review, reference 89, - Yeo KT, Oei JL, De Luca D, et al. *Review of guidelines and recommendations from 17 countries highlights the challenges that clinicians face caring for neonates born to mothers with COVID-19* [published online ahead of print, 2020 Jul 27]. *Acta Paediatr.* 2020;10.1111/apa.15495), hygiene guidelines are extremely different from one center to another and have often been changed overtime, so it is not possible to have this level of detail.

However, we totally agree with the Reviewer as this is a crucial point and late infections can be significantly reduced in case of mother-neonate rooming in if appropriate PPE and hygiene measures are used. We added some of these considerations in discussion, pag.11, 1st par.

C24. In the Discussion section

Even though from these reported cases there is a high incidence of post-natal transmission, there is no description if precautions were undertaken. There is now new published data regarding the safety of rooming-in and against mother-newborn dyad separation if precautions are undertaken. The new American Academy of Pediatrics Guidelines and a recent study published online July 23, 2020 [https://doi.org/10.1016/S2352-4642\(20\)30235-2](https://doi.org/10.1016/S2352-4642(20)30235-2) in *Lancet Child and Adolescent Health* are supporting not to separate the mother and newborns. I would revise this section of the discussion according to these new findings and recommendations and add the 2 references. As it is now, the discussion seems favoring the separation of the dyad.

A24. We agree with the Reviewer and thank for the suggestion. This part of discussion has been significantly modified (also following point C23 above – Discussion, pag.11, 1st par.), the discussion and conclusions have been toned down on this point and the *Lancet Child Health* paper has been cited, as it suggests that with the use of adequate PPE the risk of postnatal, environmentally acquired infections may be significantly reduced.

However, postnatal infections do exist and we do not know exactly if all of them happened because PPE and hygiene measures were not used or were inadequate. We do not know what are the best measures/PPE neither, and there are significant differences between AAP guidelines and those issued by other scientific societies or governmental bodies (see above and reference 88). This makes the situation quite complex and we hope to have amended enough the text to describe this.

Reviewer #3

I agree with the first reviewer regarding the assessment of study quality. (NoA: we highlighted in bold the main points here)

1. The authors have ignored major components of the systematic review. The systematic reviews aim to assess the quality of included articles to disclose the risk of bias and conclude the level of evidence.

2. The concluded level of evidence in systematic reviews is an important source for both future research and clinical recommendations. **Therefore, I suggest assessing the study quality and report the results inside the text.**

3. The statistical analysis is performed well, but the authors did not include the Forest plot. **I highly recommended the authors to include the forest plot.**

We are very grateful to the Reviewer and appreciate that the review has been given in a short time!. **We believe that there is a misunderstanding and that some points of our work might have been overlooked. We explained that in detail to the Editor and to Reviewer 1 as well (points C10-C16-C17-C20-C21). We acknowledge that the text was not enough clear, so these points may have gone unnoticed: we have rephrased some portions of the manuscript in order to highlight these missed informations.**

Please also consider the following in response to the points above highlighted in bold:

- 1. We did not ignore any component of systematic review and we strictly followed PRISMA guidelines, whose checklist is attached and has been updated** (we updated because, thanks to some useful Reviewers' comments, some page numbers changed). As ours is a meta-analysis and synthesis of case reports/series the derived level of evidence is 4, according to the Oxford Centre for Evidence-based Medicine classification. This was already commented in the Discussion, study limitations, pag.12, with reference to the classification (reference 95).
- 2. We already evaluated the study quality for each article. We did that with a specific tool dedicated to the quality evaluation of case reports/series: the Mayo Evidence-Based Practice Centre tool** (published in Murad MH et al, *BMJ Evid Based Med* 2018 – reference 22). This tool has been already used in similar meta-analyses of case reports/series (see Bazerbachi F, et al. *Gastroenterol Rep* 2017). Reviewer 1 and the Editor specifically asked to use this tool and we already used it. This is described in Methods, Assessment of risk of bias, pag.7. Results of this evaluation are reported in Tab.1 and Results, pag.8, 2nd par.

Moreover, we added, as additional analysis, two multivariate logistic regressions, with enter method, adjusting for the quality of reviewed article (evaluated by the specific tool (Mayo Evidence-Based Practice Centre - Murad MH et al, *BMJ Evid Based Med* 2018 – reference 22). The regressions had the incidence of late onset neonatal infections as dependent variable, and the mother-neonate separation or breastfeeding as independent variables. Adjusted OR were consistent with crude OR and our results did not change. This is now described in Methods, additional analyses, pag.8 and at the end of results, end of pag.9 (see also point C13 and below). This has allowed us to be extra sure about the two factors (breastfeeding and mother-neonate separation – see below) potentially influencing the incidence of late neonatal SARS-CoV-2 infections and to exclude that the quality of articles was a confounder.

- 3. We did a synthesis and meta-analysis of case reports and series. This is a less common and very different technique compared to the usual meta-analysis of RCTs or observational studies.** RCTs and observational studies report an "intervention" in a given patients' population (following randomization or not) and the effect of that intervention is evaluated by a given outcome, which is measured both in treated patients and in a comparator (i.e.: an untreated, or differently treated, patients' population). This is resumed in the so-called PICO (patient-intervention-comparator-outcome) questions acronym.
A meta-analysis of case reports/series is different because these studies usually do not have an intervention and, by definition, cannot have an external comparator. Moreover, in our case, we could not have an intervention since neonatal SARS-CoV-2 infection does not have any specifically validated or even proposed therapy. In fact, all meta-analysed case reports/series did not focus on a therapy but rather on the clinical characteristics and route of transmission of neonatal infections, which represent our main objective (please see point C8). **Thus, we did not analyse any intervention and could not have any intervention, nor an outcome potentially modified by that intervention: thus, creating a Forest plot is just impossible.**

All our knowledge on neonatal SARS-CoV-2 infections is coming from case reports/series since there are neither RCTs, nor larger observational studies, because infections in neonates are rare compared to adults. Thus, the meta-analysis of case reports and series, although less common than the meta-analysis of RCTs/observational studies is a particularly interesting technique for a relatively rare condition lacking of other studies (such as neonatal SARS-CoV-2 infection). An example of the same situation may be found in the recently published meta-analysis of case reports on the epidemiology and clinical manifestations of mucormycosis (Jeong W et al, *Clin Microbiol Infect* 2019 - doi.org/10.1016/j.cmi.2018.07.011).

We analyzed the effect of two policies (breastfeeding and mother/neonate separation, which are extremely variable across the hospitals (see our review in reference 89 - Yeo KT, Oei JL, De Luca D, et al. *Review of guidelines and recommendations from 17 countries highlights the challenges that clinicians face caring for neonates born to mothers with COVID-19 [published online ahead of print, 2020 Jul 27]. Acta Paediatr.* 2020;10.1111/apa.15495)) on the incidence of late neonatal infection. **This is the only doable analysis somehow similar to that of the classical meta-analysis of RCTs or observational studies.**

We will be anyway happy to add some of these considerations to the text or provide further modifications, if required by the Reviewer or the Editor. We also remain available to discuss new forms of data presentation if needed.